# Non-genetic inheritance restraint of cell-to-cell variation

**Harsh Vashistha[1], Maryam Kohram[1], Hanna Salman[1,2]***

[1]Department of Physics and Astronomy, The Dietrich School of Arts and Sciences, University of Pittsburgh, Pittsburgh, United States; [2]Department of Computational and Systems Biology, School of Medicine, University of Pittsburgh, Pittsburgh, United States

**Abstract** Heterogeneity in physical and functional characteristics of cells (e.g. size, cycle time, growth rate, protein concentration) proliferates within an isogenic population due to stochasticity in intracellular biochemical processes and in the distribution of resources during divisions. Conversely, it is limited in part by the inheritance of cellular components between consecutive generations. Here we introduce a new experimental method for measuring proliferation of heterogeneity in bacterial cell characteristics, based on measuring how two sister cells become different from each other over time. Our measurements provide the inheritance dynamics of different cellular properties, and the 'inertia' of cells to maintain these properties along time. We find that inheritance dynamics are property specific and can exhibit long-term memory (~10 generations) that works to restrain variation among cells. Our results can reveal mechanisms of non-genetic inheritance in bacteria and help understand how cells control their properties and heterogeneity within isogenic cell populations.

## Introduction

One of the main challenges in biological physics today is to quantitatively predict the change over time in cells' physical and functional characteristics, such as cell size, growth rate, cell-cycle time, and gene expression. All cellular characteristics are determined at all times by the interaction of genetic and non-genetic factors. While genetic information passed from generation to the next is the main scheme, by which cells conserve their characteristics, non-genetic cellular components, such as all proteins, RNA, and other chemicals, are also transferred between consecutive generations and thus influence the state of the cell's characteristics (or its phenotype) in future generations (*Lambert et al., 2014*; *Robert et al., 2010*). The mechanism of genetic information transfer between generations, as well as how this information is expressed, is mostly understood (*Casadesús and Low, 2006*; *Chen et al., 2017*; *Turnbough, 2019*). This information can be altered by rare occurring processes such as mutations, lateral gene transfer, or gene loss (*Bryant et al., 2012*; *Robert et al., 2018*). Therefore, changes resulting from genetic alterations emerge over very long timescales (several 10 s of generations). On the other hand, inheritance of non-genetic cellular components, which are subject to a considerable level of fluctuations, can influence cellular characteristics at shorter timescales (*Casadesús and Low, 2013*; *Huh and Paulsson, 2011*; *Norman et al., 2013*; *Veening et al., 2008*).

Here we focus on understanding how robust cellular characteristics are to intrinsic sources (stochastic gene expression and division noise) and extrinsic sources (environmental fluctuations) of variation, and how cells that emerge from a single mother develop distinct features and over what time scale. While our understanding of variation sources has increased significantly over the past two decades (*Ackermann, 2015*; *Avery, 2006*; *Elowitz et al., 2002*), progress in understanding non-genetic inheritance and its contribution to restraining the proliferation of heterogeneity has been extremely

**\*For correspondence:**
hsalman@pitt.edu

**Competing interests:** The authors declare that no competing interests exist.

**eLife digest** All the different forms of life on our planet – including animals, plants, fungi and bacteria – tend to grow, multiply and expand. This happens through a process called cell division, where one cell becomes two; two cells become four; four cells become eight; and so on. Each dividing cell passes on the same set of genetic instructions to its two daughter cells in the form of DNA. Its remaining contents, made up of a mixture of proteins, RNA and other chemicals, also get divided up equally between the two new cells.

This division of cellular assets establishes a form of 'cellular memory', where daughter cells retain very similar properties to their ancestors, which helps them remain stable over time. Yet this memory can fade, and small changes in how a cell looks or acts can appear over many generations of cell division. This happens even when the exact same set of DNA-based genetic instructions have been passed down to daughter cells, confirming that other factors aside from DNA do influence cellular properties and can act to maintain them or introduce variation over time.

Here, Vashistha, Kohram and Salman set out to understand how long cellular memory could be maintained in dividing *E. coli* bacteria. To do this, they created a technique to track cellular memory as it passed down from a single mother cell to two daughter cells over dozens of generations. Using this technique, Vashistha, Kohram and Salman found that some inherited elements, including cell size and the time cells took to divide, were maintained between mother and daughter cells for almost 10 generations. Other elements, such as the density of proteins inside each cell, started changing almost immediately after daughter cells were formed, and only remained similar for about two generations.

These findings suggest that cellular memory may be long, but is not infinite, and that inheritance of non-genetic elements can help maintain cellular memory and reduce variation among new-born cells for considerable number of generations. Building on this research to achieve a better understanding of cellular memory may allow researchers to harness these insights to direct the evolution of different cellular properties over time. This could have a wide range of potential applications, such as designing new infection control measures for viruses or bacteria; enhancing our ability to grow working organs for tissue transplant; or improving the texture and consistency of cultured, lab-grown meat.

limited. Extensive studies have been dedicated to revealing the different non-genetic mechanisms that influence specific cellular processes and how they are inherited over time (*Chai et al., 2010*; *Govers et al., 2017*; *Mosheiff et al., 2018*; *Sandler et al., 2015*; *Wakamoto et al., 2005*). However, the cell's phenotype is determined by the integration of multiple processes. Thus, to predict the inheritance dynamics of a cellular phenotype, we need to measure the inheritance dynamics directly rather than characterizing the effect of individual inheritance mechanisms separately. Progress in this research has been drastically hindered by the limited experimental techniques that can provide reliable quantitative measurements.

The recent development of the 'mother machine' (*Brenner et al., 2015*; *Wang et al., 2010*) has provided valuable data of growth and division, as well as protein expression dynamics. These data have been used to gain insight into non-genetic inheritance and cellular memory. The results obtained have consistently showed that non-genetic memory in bacteria is almost completely erased within two generation (*Susman et al., 2018*; *Tanouchi et al., 2015*; *Wang et al., 2010*). This has also been the conclusion of theoretical calculations of cell size autocorrelation (*Ho et al., 2018*; *Susman et al., 2018*), which are based on the adder model (*Amir, 2014*; *Taheri-Araghi et al., 2015*; *Si et al., 2019*) for size homeostasis. The consensus of previous experimental studies is founded on the calculation of the autocorrelation function (ACF) for the different measurable cellular properties, such as cell size, growth rate, cell-cycle time, and protein content. It is important to note that in calculating the ACF, measurements of cells from different traps of the mother machine are averaged together. However, small variation in the trap sizes can manifest during the fabrication process, which can lead to distinct environments in different traps (*Yang et al., 2018*). In addition, cells might experience slightly different environments at different times resulting from thermal fluctuations and their dynamic interaction with their surroundings, i.e. environmental fluctuations can influence the

cell's growth and division dynamics, which in turn can change the cell's micro-environment through consumption of nutrients and/or secretion of other chemicals. As a result of the individuality of the cell–environment interaction, different micro-niches can be created in different traps (as we demonstrate later in the Results section), which give rise to diverse patterns of growth and division dynamics and therefore distinct ACFs (*Figure 1—figure supplement 1*) (see also *Susman et al., 2018*; *Yang et al., 2018*; *Tanouchi et al., 2015*). Averaging over many traps, with such various ACFs, will thus erase the dynamics of cellular memory.

To overcome this hurdle, we have developed a new measurement technique, which enables us to separate environmental effects from cellular ones. The technique is based on a new microfluidic device that allows trapping two cells immediately after they divide from a single mother simultaneously and sustain them right next to each other for extended time. Thus, with this technique, we track the lineages of the two sister cells (SCs) from the time of their birth and follow them as they age together for tens of generations. This enables us to measure how two cells that originate from the same mother become different over time, while experiencing exactly the same environment. Thus, we are able to measure the non-genetic memory of bacterial cells for several different traits. Our results reveal important features of cellular memory. We find that different traits of the cell exhibit different memory patterns with distinct timescales. While the cell-cycle time and cell size exhibit slow exponential decay of their memory that extends over several generations, other cellular features exhibit complex memory dynamics over time. The growth rates of two SCs, for example, diverge immediately after division, but re-converge toward the end of the first cell cycle and subsequently persist together for several generations. In comparison, the mean fluorescence intensities, reporting gene expression, are identical in both cells immediately after they separate but diverge within two cell cycles.

## Results

Our new microfluidic device, dubbed the 'sisters machine' (*Figure 1A*), consists of 30 μm long narrow trapping channels (1 μm—1 μm) open at one end to a wide channel (30 μm—30 μm), through which fresh medium is continuously pumped to supply nutrients to cells in the traps and wash away cells that are pushed out of them. Here however, every two neighboring trapping channels are joined on the closed end through a v-shaped connection of the same width and height. The tip of the v-shaped connection is made 0.5 μm narrower than the rest of the channel to reduce the likelihood of cells passing from one side to the other (*Figure 1B*). Therefore, once it happens, the cells at the tip will remain there, while we track their growth and division events, and measure their size and protein expression (*Figure 1C,D*), until the next cell passage occurs, which can take 10 s of generations (see *Figure 1—video 1*). The environment in this setup is identical for both cells at the tip of the v-shaped connection, as they are kept in close proximity to each other. This ensures that differences observed between the two cells are due to internal cellular factors only. A comparison of the growth patterns of two pairs of SCs measured in the same experiment, where each pair shares a common trap, reveals that while the growth dynamics of SCs are strikingly similar, they are significantly different between the two pairs (*Figure 2A*). This is further confirmed by comparing the distribution of the difference between the average growth rates of SCs to that of pairs of cells residing in different channels (*Figure 2B*). These results highlight the significance of the contribution of environmental fluctuations to cellular growth dynamics and support the existence of different environmental micro-niches within our, and similar, microfluidic setups as mentioned earlier. Note, however, that cell division in the new v-shaped channels does not alter the statistics of SCs' relative sizes, growth rates, or generation times, in comparison to that observed in the case of division in straight channels (*Figure 3*).

Using this setup, we successfully trapped pairs of cells next to each other for $20-160$ generations. Images of the cells in both Differential Interference Contrast (DIC) and fluorescence modes were acquired every 3 min. Under our experimental conditions (cells growing in LB medium at 32°C), the average generation time was $34 \pm 7$ min, which provided ~11 images every generation. The acquired images were used to measure various cellular characteristics as a function of time, including cell size, protein concentration, growth rate, and generation time. To measure cellular memory, we replace the ACF, used in previous studies, with the Pearson correlation function (PCF) between pairs of cells:

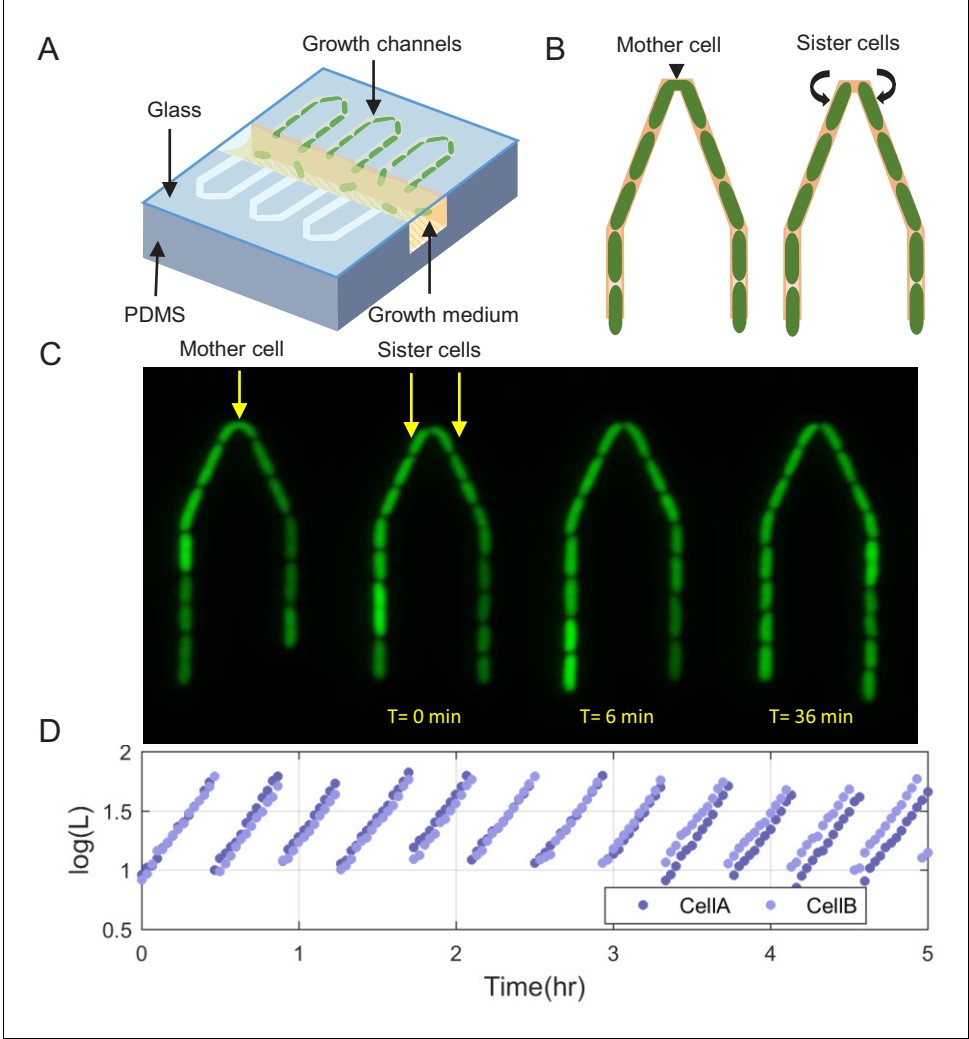

**Figure 1.** Scheme of the experimental setup for tracking sister cells. (**A**) Long (30 μm) narrow traps (1 μm—1 μm) are connected on one end and open on the other to wide (30 μm—30 μm) perpendicular flow channels through which fresh medium is pumped and washes out cells that are pushed out of the traps. (**B**) Illustration of SCs being born from a single mother cell at the tip of the trap, as can also be seen in real fluorescence images of the cells in the trap (**C**), which are then followed for a long time (see **Figure 1—video 1**). (**D**) Section of example traces of two sister cells from the time they are born, which shows how they become different over time.

The online version of this article includes the following video and figure supplement(s) for figure 1:

**Figure supplement 1.** The ACFs of individual lineages measured in separate traps.

**Figure 1—video 1.** Creation of sister cells (SCs) in the experimental setup.

https://elifesciences.org/articles/64779#fig1video1

$$PCF^{(y)}(t) = \frac{1}{\sigma_{y^{(1)}} \sigma_{y^{(2)}}} \sum_{i=1}^{n} (y_i^{(1)}(t) - <y^{(1)}>).(y_i^{(2)}(t) - <y^{(2)}>) \qquad (1)$$

where y is the cellular property of interest, t is the measurement time, n is the number of cell pairs measured, $\sigma_y$ is the population standard deviation of y, and (1) and (2) represent the two cells being considered. $PCF^{(y)}(t)$ is therefore a measure of the correlation between the values of a specific cellular property at time t. We use this correlation function to compare three types of cell pairs (**Figure 4A**): (1) SCs are cells that originate from the same mother at time 0, and therefore, the value of PCF at time 0 is 1. (2) Neighbor cells (NCs) are cells that reside next to each other at the tip of the v-shaped connection. However, NCs are cells that do not originate from the same mother. They

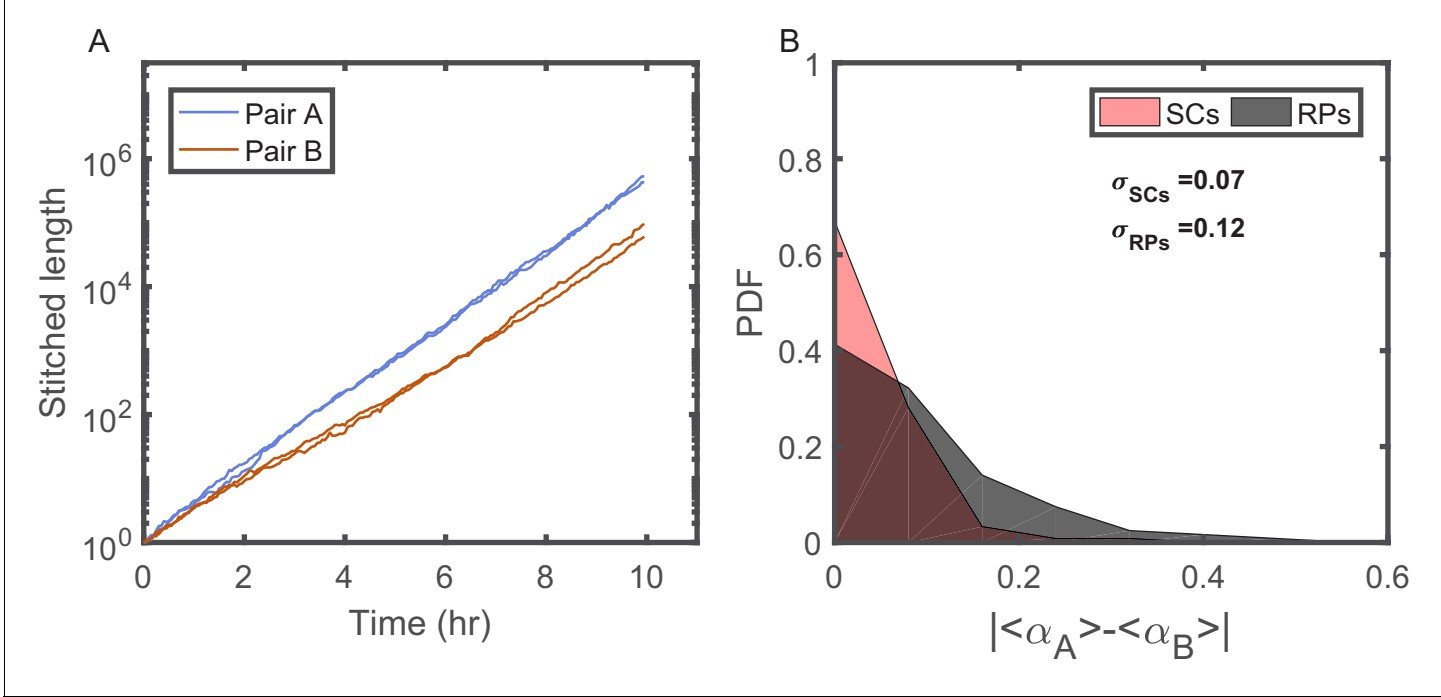

**Figure 2.** Individuality of cellular growth dynamics in different microenvironments. (**A**) Depicts the cell length of two pairs of SCs measured in two different V-shaped traps as a function of time. The length of each cell is presented in a 'stitched' form, where the length of the cell in each cell cycle is adjusted to start from the length of the cell at the end of the previous cycle, ignoring by this the division events. This is done by dividing the length in each cycle by the starting length and multiplying it by the length of the cell at the end of the previous cycle. This presentation emphasizes the difference in the average growth rates measured in different traps. Note, however, that each pair of SCs exhibits similar average growth rate. (**B**) Probability distribution function (PDF) of the absolute difference in the average growth rate of two SCs is compared with the absolute difference in the average growth rate of two randomly paired cells (RPs) growing in separate traps in the same device (see *Figure 4A* for further elaboration on random pairing of cells). The standard deviation of the difference for SCs ($\sigma_{SCs}$) is almost half of the calculated value for RPs ($\sigma_{RPs}$). This shows that cells grow with different average growth rates in different traps and supports the idea of micro-niche formation in the microfluidic device.

are cells that happen to enter into both sides of the same v-shaped channel from the start of the experiment. We initiate their tracking though, only when they happen to divide at the same time, such that at time 0, they are both at the start of a new cell cycle, and if their length is almost identical at that point in time. This choice is to ensure that any long-term correlation measured in SCs does not stem from a size homeostasis mechanism, which would maintain the size of both cells similar for several generations if they start similar. (3) Random cell pairs (RPs) are cells that reside in different traps and their lineages are aligned artificially even though they can be measured at different times. In this case, t is measured relative to the alignment point, which is chosen to be at the start of the cell cycle for both cells. Since NCs and RPs do not originate from the same mother at time 0, the PCF is measured from the first generation only, and we set it to be 1 at time 0. Comparing the correlation of NCs, which experience the same environmental conditions at the same time, with that of RPs allows us to determine the effect of the environment on the correlation. On the other hand, the comparison of SCs with NCs provides the effect of cellular factors (i.e. epigenetics) that are shared between SCs, on the correlation function. This in turn allows us to determine the cellular memory of a specific property resulting from shared information passed on from the mother to the two sisters (see Appendix for the mathematical relationship between the different measures).

We measured the correlations between the different pair types for cell-cycle time (T). We find that T of SCs remain strongly correlated for up to eight successive cell divisions (*Figure 4B* also see *Figure 4—figure supplements 1* and *2*) regardless of the environmental conditions (*Figure 4—figure supplement 3*), while the NCs correlation decays to zero within three generations (*Figure 4C*). These results clearly reveal the effects of epigenetics and environmental conditions on cellular memory when compared to the RPs correlation, which as expected decays to zero within one generation similar to the ACF (*Figure 4B,C*).

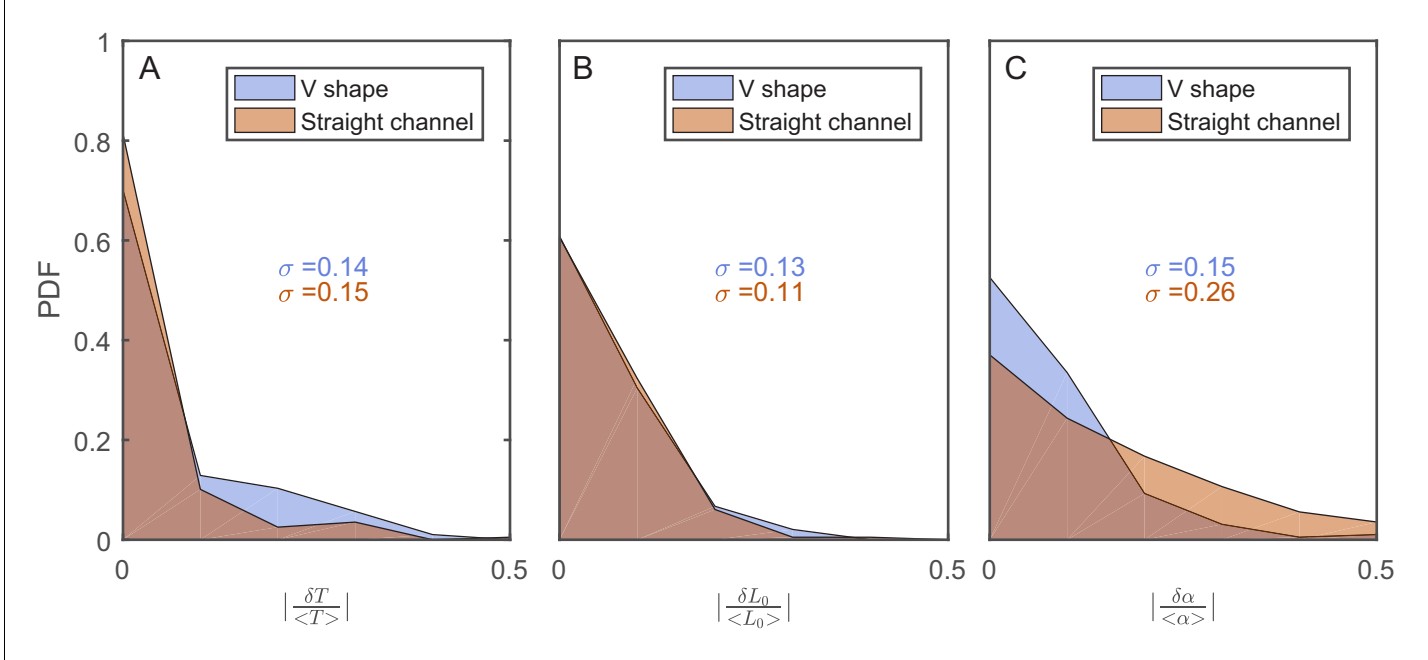

**Figure 3.** The effect of the v-shaped channel on the distribution of the different cellular characteristics between SCs during division. (**A**) Probability distribution Function (PDF) of the difference in the first cell-cycle time of two sister cells after separation relative to the population's average cycle time under the same experimental conditions. (**B**) PDF of the difference in cell length between the sister cells immediately after division relative to the population's average length at the start of the cell cycle. (**C**) PDF of the difference in the growth rate of the two sister cells after separation relative to the population's average growth rate. The difference measured in the straight channels here is larger than that measured in the v-shaped channels. This could be due to the fact that the two cells in the mother machine trap are at different distance from the nutrients diffusing from the flow channel into the traps. This has been shown before to result in variation in the cells growth rate (**Yang et al., 2018**). In all graphs, the blue curves represent the distributions measured in our new device with the v-shaped channels using 194 pairs, while the brown curves were measured in the straight channels of the mother machine using 198 pairs.

Next, we applied our method to cell size. Also here, our measurements show that SCs correlation decays slowly over ~7 generations (**Figure 4D**), while the correlation of NCs exhibit fast decay to zero within two generations similar to the ACF (**Figure 4E**). Note that RPs exhibit no correlation from the start of the measurement (**Figure 4D,E**). These results further demonstrate the existence of strong non-genetic memory that restrains the variability of cell size between SCs for a long time. Unlike the cell-cycle time however, the effect of both epigenetic factors and environmental conditions on the cellular memory appears to extend for a slightly shorter time.

To quantify the increase in variability among cells along time differently, we measured the change in the variance of a cellular property as time advances, which is expected to reach an equilibrium saturation value at long timescales. Measuring how the variance reaches saturation provides information about cellular memory and the nature of forces acting to restrain variation. The cellular memories of cell-cycle time and length, measured using this method, agree well with our previous PCF results (**Figure 5—figure supplements 1** and **2**). Thus, we have measured the relative fluctuations in the exponential elongation rate of the cell pairs $\delta\alpha$ defined as:

$$\delta\alpha(t) = \alpha^{(1)}(t) - \alpha^{(2)}(t) \tag{2}$$

where $\alpha(t) = (d \ln L / dt)$ is the exponential elongation rate of the cell, L(t) is the cell length at time t, and (1) and (2) distinguish the cell pair (**Figure 5—figure supplement 3**). As expected, $\delta\alpha$ for all pairs of lineages is randomly distributed with $\langle \delta\alpha \rangle = 0$ (**Figure 5—figure supplement 3**), as the elongation rate of all cells fluctuate about a fixed value identical for all cells in the population and depends on the experimental conditions. The variance of $\delta\alpha$ for both RPs ($\sigma^2_{\delta\alpha_{RPs}}$) and NCs ($\sigma^2_{\delta\alpha_{NCs}}$) was found to be constant over time and is similar for both types of cell pairs (**Figure 5A**). However, the variance of $\delta\alpha$ for SCs ($\sigma^2_{\delta\alpha_{SCs}}$) exhibits a complex pattern (**Figure 5B**), which eventually converges

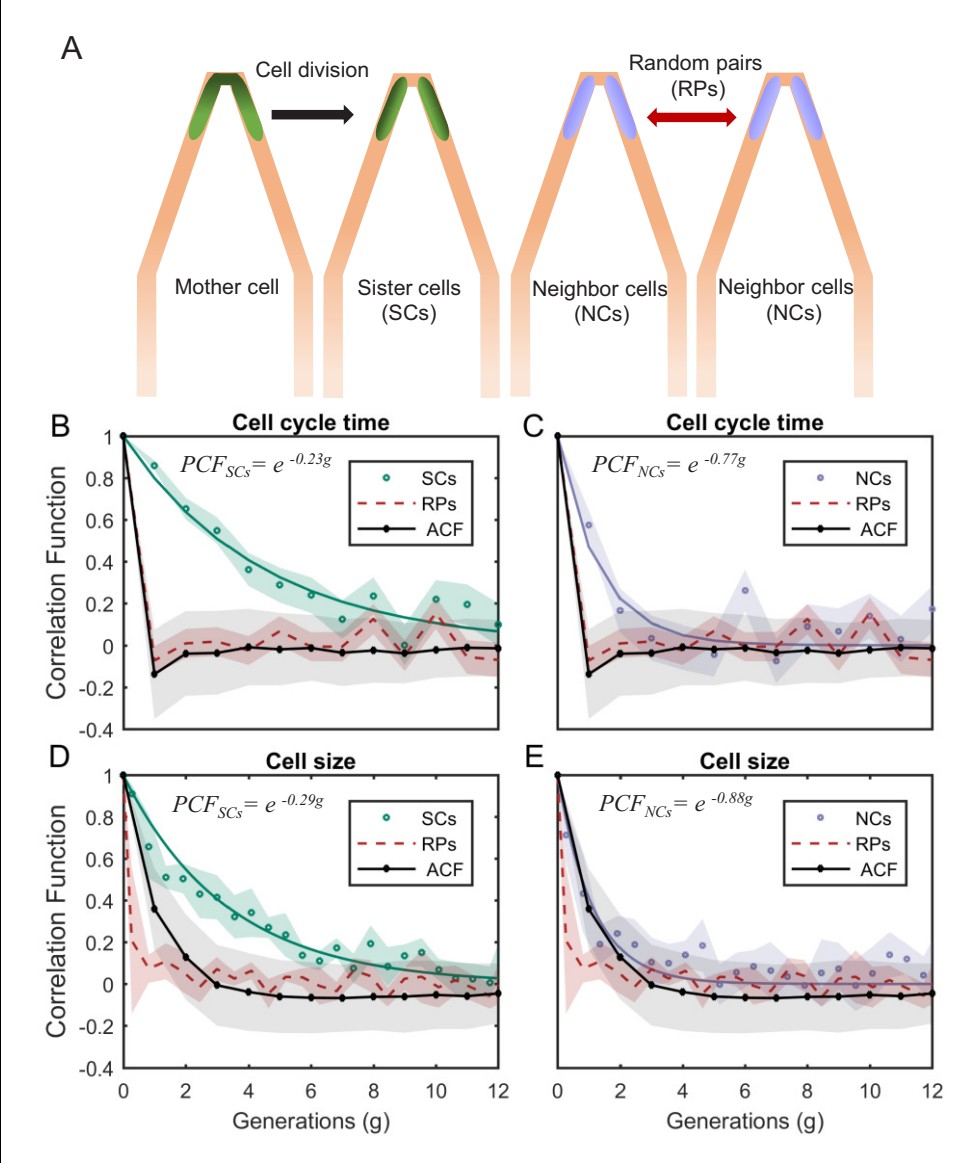

**Figure 4.** PCF of cell-cycle time and cell size measured in cell pairs as a function of number of generations. (**A**) Three types of pairs used for calculating PCF. (**B**) PCF of cell-cycle time for SCs (122 pairs from three separate experiments) exhibit memory that extends for almost nine generations (half lifetime ~ 4.5 generations). This is ~3.5× longer than the half lifetime of NCs PCF (calculated using a 100 pairs from three separate experiments) (**C**), which is comparable to the ACF (half lifetime ~1 generation). (**D**) Similarly, SCs exhibit strong cell size correlation that decays slowly over a long time (half lifetime ~3.5 generations), while (**E**) NCs show almost no correlation in cell size similar to ACF of initial sizes (half lifetime ~1 generation). For details of the cell-cycle time PCF and errors calculation see SI and *Figure 4—figure supplements 1* and *2*. PCF values for cell size were calculated in similar way to cell-cycle time and were then averaged over a window of six consecutive time frames (15 min time window) (See *Figure 4—figure supplement 4* for raw data). Shaded area represents the standard deviation of the average. The equations in the graphs represent the best fit to the PCF depicted in each graph with g is generation number. The online version of this article includes the following figure supplement(s) for figure 4:

**Figure supplement 1.** Distributions of different cell parameters.

**Figure supplement 2.** Correlation in cell-cycle times for SCs was verified by calculating slopes of best fits to the plots of normalized TimeA vs TimeB.

**Figure supplement 3.** The PCF of cell-cycle time (T) for SCs in different growth conditions.

**Figure supplement 4.** Raw PCF values of cell size as a function of time for SCs, NCs, and RPs.

**Figure supplement 5.** PCF values of cell size and cell-cycle duration as a function of time for NCs with different starting sizes.

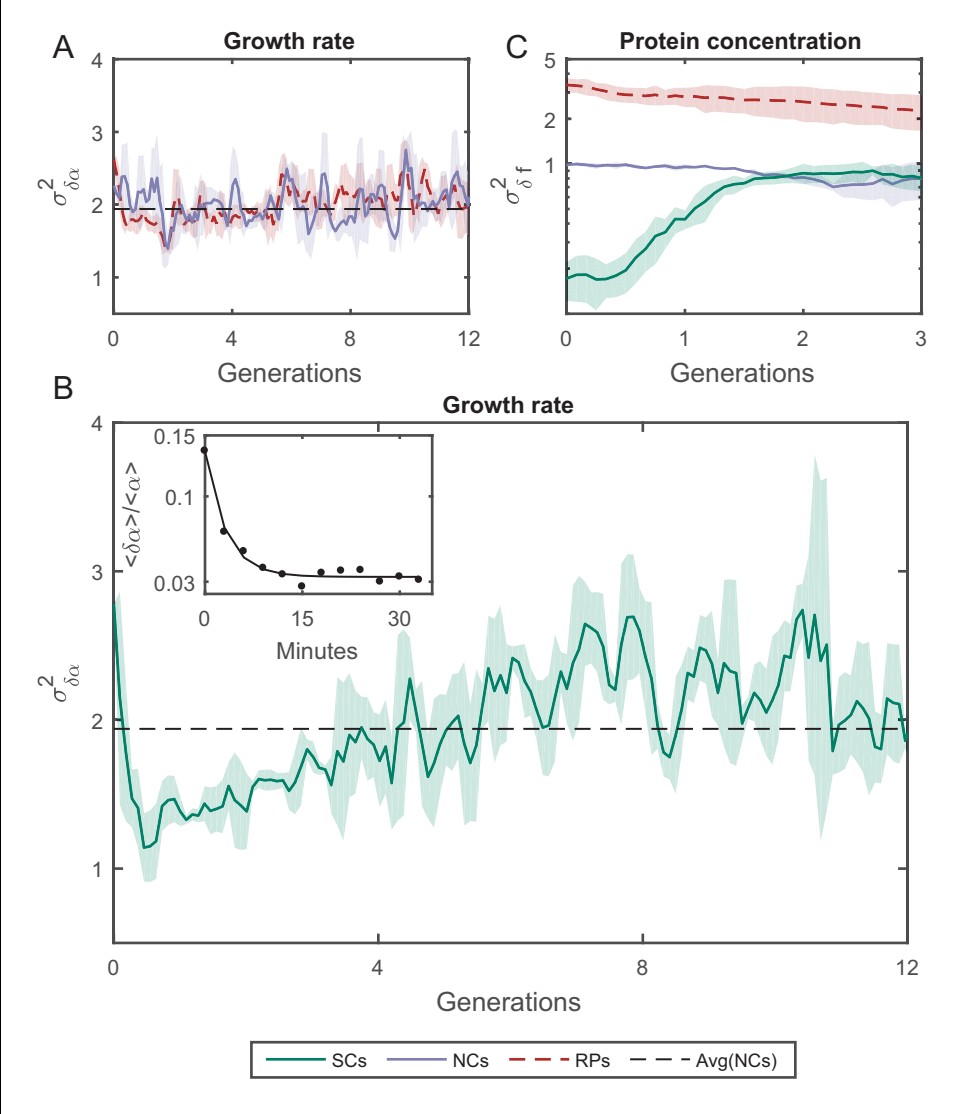

**Figure 5.** Variance ($\sigma^2_{\delta\alpha}$) as a function of the time. (A) $\sigma^2$ of the growth rate difference ($\delta\alpha$) between cell pairs for NCs and RPs as a function of time (see *Figure 5—figure supplement 3* for the details of the calculation). The variance for both pair types does not change over time. (B) $\delta\alpha$ of SCs, on the other hand, exhibits large variance immediately after separation (~50%) higher than NCs and RPs and rapidly drops to its minimum value within one generation time (~30 min), and increases thereafter for 4 hr (~8 generations) until saturating at a fixed value equivalent to that observed for NCs and RPs. Each point in A and B is the average over three frames moving window, and the shaded area represents the standard deviation of that average. (C) Unlike $\delta\alpha$, $\delta f$ of SCs increases to its saturation value within ~2 generations (see *Figure 5—figure supplement 4* for the details of the calculation). Here, each point represents the average of three different experiments, and the shaded part represents the standard deviation.

The online version of this article includes the following figure supplement(s) for figure 5:

**Figure supplement 1.** Cell-cycle time variance ($\sigma^2_{\delta T}$) as a function of time.

**Figure supplement 2.** Cell size variance ($\sigma^2_{\delta L_0}$) as a function of time.

**Figure supplement 3.** Exponential elongation rate difference ($\delta\alpha$) as a function of time.

**Figure supplement 4.** Mean fluorescence variance ($\sigma^2_{\delta f}$) as a function of time.

to the same value as RPs ($\sigma^2_{\delta\alpha_{RPs}}$) and NCs ($\sigma^2_{\delta\alpha_{NCs}}$). The time it takes for ($\sigma^2_{\delta\alpha_{SCs}}$) to reach saturation extends over almost eight generations, which again reflects a long memory resulting from epigenetic factors. These results show that, unlike cell-cycle time and cell length, elongation rates of SCs immediately after their division from a single mother exhibit the largest variation. This variation decreases to its minimum value within a single cell-cycle time (~30 min). To understand the source of this large variation immediately following separation, we have measured the growth rate over a moving time window of 6 min throughout the cell cycle and compared the results between SCs. Our comparison clearly shows that an SC that receives a smaller size-fraction from its mother exhibits a larger growth rate immediately after division. The growth rate difference between the small and large sisters decreases to almost zero by the end of the first cell cycle after separation (*Figure 5B* inset). This result reveals that the exponential growth rate of a cell immediately after division inversely scales with the size-fraction the cell receives from its mother (see also *Kohram et al., 2020*). It also demonstrates that the difference in the growth rates between SCs changes during the cell cycle, indicating that they are not constant throughout the whole cycle as has been accepted so far (*Godin et al., 2010*; *Soifer et al., 2016*; *Wang et al., 2010*). Note that similar results have been reported recently for *Bacillus subtilis* (*Nordholt et al., 2020*), where it was observed that the growth rate is inversely proportional to the cell size at the start of the cell cycle and changes as the cell-cycle advances.

We have also examined how the protein concentration varies over time between the two cells by measuring the concentration of GFP (green fluorescent protein), via its fluorescence intensity, expressed from a constitutive promoter in a medium copy-number plasmid. The variance of fluorescence intensity difference between cell pairs δf was calculated as for the growth rate (see *Figure 5—figure supplement 4* for details). Upon division, soluble proteins are partitioned symmetrically with both daughters receiving almost the same protein concentration. As expected, $\sigma^2_{\delta f_{SCs}}$ starts from ~0 initially, and diverges to reach saturation within two generations (*Figure 5C*). On the other hand, NCs and RPs exhibit constant variance throughout the whole time, with $\sigma^2_{\delta f_{RPs}}$ twice as large as $\sigma^2_{\delta f_{NCs}}$, which reflects the influence of the shared environment resulting in additional correlations between NCs. The relatively short-term memory in protein concentration may be protein specific (*Figure 5—figure supplement 4*), or it could reflect the fact that in this case the protein is expressed from a plasmid. Nevertheless, this result indicates that cellular properties are controlled differently and can exhibit distinct memory patterns. It is important therefore to distinguish between different cellular characteristics and to examine their inheritance patterns individually.

## Discussion

There has been a rising interest over the past two decades in understanding the contribution of epigenetic factors to cellular properties and their evolution over time. Here, we introduce a new measurement technique that can separate environmental fluctuations from cellular processes. This allows for quantitative measurement of non-genetic memory in bacteria and reveals its contribution to restraining the variability of cellular properties. Our results show that the restraining force dynamics vary significantly among different cellular properties, and its effects can extend up to $sim_{10}$ generations. In addition, the growth rate variation emphasizes the effect of division asymmetry, which can help in understanding the mechanism that controls cellular growth rate. The slow increase in the growth rate variance that follows reflects the effect of inheritance. Since both cells inherit similar content, which ultimately determines the rate of all biochemical activities in the cell and thus its growth rate, it is expected that both cells would exhibit similar growth rates once they make up for the uneven partitioning of size acquired during division. The short memory we see in the protein concentration, on the other hand, suggests that cells are less restrictive of their protein concentration. This might be protein specific, or for proteins that are expressed from plasmids only. Nevertheless, these results highlight the importance of such studies, and how this new method can help answer fundamental questions about non-genetic memory and variability in cellular properties.

Finally, in order to understand and characterize the evolution of population growth rate as it reflects its fitness, there is a need to incorporate inheritance effects, which has been thus far assumed to be short lived. This study confirms that cellular memory can persist for several generations, and therefore limits the variation in certain cellular characteristics, including growth rate. Such

memory should be considered in future studies and has the potential of changing our perception of population growth and fitness.

# Materials and methods

## Key resources table

| Reagent type (species) or resource | Designation | Source or reference | Identifiers | Additional information |
|---|---|---|---|---|
| Strain, strain background (*Escherichia coli*) | MG1655 | Coli Genetic Stock Center (CGSC) | 6300 | F-, λ−, rph-1 |
| Recombinant DNA reagent | pZA3R-GFP | *Lutz and Bujard, 1997* | https://academic.oup.com/nar/article/25/6/1203/1197243 | GFP expressed from the λ Pr promoter |
| Recombinant DNA reagent | pZA32wt-GFP | *Lutz and Bujard, 1997* | https://academic.oup.com/nar/article/25/6/1203/1197243 | GFP expressed from the LacO promoter |
| Software, algorithm | MATLAB | MathWorks | N/A | |
| Software, algorithm | Oufti | *Paintdakhi et al., 2016* | http://oufti.org/ | |

## Device fabrication

The master mold of the microfluidic device was fabricated in two layers. Initially, the growth channels for the cells were printed on a 1 mm × 1 mm fused silica substrate using Nanoscribe Photonic professional (GT). The second layer, containing the main flow channels that supply nutrients and wash out excess cells, was formed using standard soft lithography techniques (*Jenkins, 2013*; *Martinez-Duarte and Madou, 2016*). SU8 2015 photoresist (MicroChem, Newton, MA) was spin coated onto the substrate to achieve a layer thickness of 30 μm and cured using maskless aligner MLA100 (Heidelberg Instruments). Following a wash step with SU8 developer, the master mold was baked and salinized. The experimental setup described in the main text was then prepared using this master mold, from PDMS prepolymer and its curing agent (Sylgard 184, Dow Corning) as described in previous studies.

## Cell culture preparation

The wild-type MG1655 *E. coli* bacteria were used in all experiments described. Protein content was measured through the fluorescence intensity of green fluorescent protein (GFP) inserted into the bacteria on the medium copy-number plasmid pZA (*Lutz and Bujard, 1997*). The expression of GFP was controlled by one of two different promoters, the Lac Operon (LacO) promoter was used to measure the expression level of a metabolically relevant protein, while the viral λ-phage Pr promoter was used to measure the expression level of a constitutive metabolically irrelevant protein.

Two testing media were used in our experiments. M9 minimal medium supplemented with 1 g/l casamino acids and 4 g/l lactose (M9CL) was used for measuring the expression level from the LacO Promoter, and LB medium was used for all other experiments. The cultures were grown over night at 30˚C, in either LB or M9CL medium depending on the intended conditions. The following day, the cells were diluted in the same medium and regrown to early exponential phase, optical density (OD) between 0.1 and 0.2. When the cells reached the desired OD, they were concentrated into fresh testing medium to an OD~0.3 and loaded into a microfluidic device. Once enough cells were trapped in the channels, fresh testing medium was pumped through the wide channels of the device to supply the trapped cells with nutrients and wash out extra cells that are pushed out of the channels. The cells were allowed to grow in this device for days, while maintaining the temperature, using a microscope top incubator (Okolab, H201-1-T-UNIT-BL).

## Image acquisition and data analysis

Images of the channels were acquired every 3 min (in LB medium) or 7 min (in M9CL medium) in DIC and fluorescence modes using a Nikon eclipse Ti2 microscope with a 100× objective. The size and protein content of the SCs were measured from these images using the image analysis software Oufti (*Paintdakhi et al., 2016*). The data were then used to generate traces such as in *Figure 1D*,

and for further analysis as detailed in the main text. Single-cell measurements were analyzed using MATLAB. Sample ACFs, Pearson correlation coefficients, sample distributions, and curve fitting were all calculated by their implementations in MATLAB toolboxes.

## Acknowledgements

We thank Naama Brenner for helpful discussions and comments on the manuscript. This work was supported by the National Science Foundation (Grant No. Phy-**2014116**), and the US-Israel Binational Science Foundation (Grant No. 2016376).

## Additional information

### Funding

| Funder | Grant reference number | Author |
|---|---|---|
| United States-Israel Binational Science Foundation | 2016376 | Hanna Salman |
| National Science Foundation | 2014116 | Hanna Salman |

The funders had no role in study design, data collection and interpretation, or the decision to submit the work for publication.

### Author contributions

Harsh Vashistha, Resources, Data curation, Software, Formal analysis, Validation, Investigation, Methodology, Writing - review and editing; Maryam Kohram, Data curation, Investigation, Methodology; Hanna Salman, Conceptualization, Supervision, Funding acquisition, Investigation, Methodology, Writing - original draft, Project administration, Writing - review and editing

### Author ORCIDs

Harsh Vashistha (ID) https://orcid.org/0000-0002-9411-5137
Hanna Salman (ID) https://orcid.org/0000-0002-5847-524X

### Decision letter and Author response

Decision letter https://doi.org/10.7554/eLife.64779.sa1
Author response https://doi.org/10.7554/eLife.64779.sa2

## Additional files

### Supplementary files

• Transparent reporting form

### Data availability

All data generated or analyzed during this study, are available on Zenodo at http://doi.org/10.5281/zenodo.4476617.

The following dataset was generated:

| Author(s) | Year | Dataset title | Dataset URL | Database and Identifier |
|---|---|---|---|---|
| Vashistha H, Kohram M, Salman H | 2021 | Non-genetic inheritance restraint of cell-to-cell variation | http://doi.org/10.5281/zenodo.4476617 | Zenodo, 10.5281/zenodo.4476617 |

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

## Appendix 1

### Mathematical framework

Assuming that $x(t)$ is a measurable cellular property, such as cells size, growth rate, etc. We can present it as:

$$x(t) = x + \delta x(t)$$

where $x$ is the average of $x(t)$ over time, and $\delta x$ is its fluctuations around $x$. The difference of this measured property between two cells:

$$\Delta x(t) = x_1(t) - x_2(t) \tag{4}$$

where 1 and 2 represent the two different cells, will average to zero, that is, $<\Delta x> = 0$. Its variance on the other hand will be:

$$\sigma_{\Delta x}^2(t) = <\Delta x(t)^2> - <\Delta x(t)>^2 = 2<\delta x^2(t)> - 2<\delta x_1(t)\delta x_2(t)> \tag{5}$$

where $<\delta x^2> = <\delta x_1{}^2> = <\delta x_2{}^2>$ is the variance of $x$, which is the same for all cells, and $<\delta x_1(t)\delta x_2(t)>$ is the covariance of the fluctuations in both cells, which when normalized by $\sigma_{\delta x_1} . \sigma_{\delta x_2}$ would give the correlation, that is, the PCF, between the two variables. On the other hand, if we assume that $x$ is determined by two factors, internal cellular composition ($I(t)$) and external environmental conditions ($E(t)$), such that:

$$x(t) = I(t) + E(t) \tag{6}$$

Then $\sigma_{\Delta x}^2(t) = <[(I_1 - I_2) + (E_1 - E_2)]^2>$ would depend on whether the two cells share the same environment and/or the same cellular compositions. Therefore, random pair of cells (RPs), which reside in different channels and thus do not share neither the environment nor the internal composition would exhibit a variance:

$$RPs: \sigma_{\Delta x}^2(t) = 2\sigma_I^2 + 2\sigma_E^2 + 4cov(I,E) \tag{7}$$

where $\sigma_I^2 = <I^2> - <I>^2$ is the variance in the internal composition of the cell (similar for all cells and constant over time), $\sigma_E^2 = <E^2> - <E>^2$ is the variance in the environmental conditions (also the same for all cells in the same experiment), and $cov(I,E)$ is the covariance of the environment and the internal composition of the cell, which as discussed earlier can influence each other in a trap-specific manner. However, averaging many measurements from different traps erases this effect as clear from *Figure 1—figure supplement 1* (see also *Susman et al., 2018*). On the other hand, for cells that share the environment but not their internal composition, that is, neighboring cells (NCs), the variance would be:

$$NCs: \sigma_{\Delta x}^2(t) = 2\sigma_I^2 \tag{8}$$

Note that when the NCs are chosen to have similar size and divide simultaneously at time zero, this variance for cell size would be small initially and its increase would not be constrained by the epigenetic similarity between the two cells as in the case of sister cells (SCs). And finally, for SCs, which share both the environment and their internal composition, which means that $I_1$ and $I_2$ can be correlated, then:

$$SCs: \sigma_{\Delta x}^2(t) = 2\sigma_I^2 - 2cov(I_1,I_2) \tag{9}$$

where $cov(I_1,I_2)$ is the covariance of the internal states of the cells as a function of time, that is, the non-genetic memory of the cell. Using the definitions above, it is easy to see the relationship between the variance and the PCF. It is also clear that the difference between NCs and RPs variances would provide the contribution of the environment, while the difference between SCs and NCs

variances would give the contribution of the internal composition of the cell to the variance, or the epigenetic memory.

**Appendix 1—table 1.** The calculated values of the PCF for SCs were verified by calculating the slopes of best fits to the plots of TimeA vs TimeB graphs (***Figure 4—figure supplement 2***).

| Generation | PCF $\pm \sigma_{PCF}$ | Slope of best fit line (*Figure 4—figure supplement 2*) |
|---|---|---|
| 1st | 0.86 ± 0.02 | 0.87 |
| 2nd | 0.65 ± 0.05 | 0.69 |
| 3rd | 0.54 ± 0.06 | 0.44 |
| 4th | 0.36 ± 0.07 | 0.42 |
| 5th | 0.28 ± 0.08 | 0.25 |
| 6th | 0.23 ± 0.08 | 0.25 |
| 7th | 0.12 ± 0.09 | 0.11 |
| 8th | 0.23 ± 0.09 | 0.25 |
| 9th | 0.00 ± 0.09 | 0.00 |

## Appendix 2

### Supplementary material

PCF and error calculation

The PCF was calculated using following equation:

$$PCF^{(y)}(t) = \frac{1}{\sigma_{y^{(1)}}\sigma_{y^{(2)}}} \sum_{i=1}^{n} (y_i^{(1)}(t) - <y^{(1)}>).(y_i^{(2)}(t) - <y^{(2)}>) \tag{3}$$

and the standard deviation (**Bowley, 1928**):

$$\sigma_{PCF} = \frac{(1 - PCF^2)}{\sqrt{n}} \tag{4}$$

where n is the number of cell pairs considered in the calculation.

