## [Decision Letter]

[Editors’ note: the authors submitted for reconsideration following the decision after peer review. What follows is the decision letter after the first round of review.]

Thank you for submitting your work entitled "Non-genetic inheritance restraint of cell-to-cell variation" for consideration by *eLife*. Your article has been reviewed by three peer reviewers, and the evaluation has been overseen by a Reviewing Editor and a Senior Editor. The following individual involved in review of your submission has agreed to reveal their identity: Minsu Kim (Reviewer #2).

Our decision has been reached after consultation between the reviewers. Based on these discussions and the individual reviews below, we regret to inform you that your work will not be considered further for publication in *eLife* at this time. However, if you feel you can satisfy the reviewers' comments, we encourage you to resubmit.

As you will see from the reviews which are copied below, all three reviewers as well as the Reviewing Editor were enthusiastic about your innovative microfluidic device and its ability to assess sister cell development and the impact of microenvironment on cell fate. However, in addition to specific concerns listed below, there was consensus that the data themselves represent a more modest advance over previous work. This was unfortunate, as reviewer 2 noted, as the set-up has the potential to permit disentangling of the effect of micro-environment and inheritance, a thorny and important problem that has not yet been solved.

Given the dichotomy of opinion about the potential impact of the approach versus the included results, we would encourage you to consider obtaining additional, data specifically addressing the relative effect of microenvironment and inheritance on cell fate. If you are amenable to this idea, *eLife* would be happy to consider the manuscript as a new submission at a later date. At the same time, we recognize that obtaining additional data is time consuming and completely understand if you would prefer to publish elsewhere.

Reviewer #2:

This article describes correlation timescale of various cellular properties. The experimental design is ingenious, and findings shed new light on epigenetic inheritance. I believe that this article warrants publication in *eLife*.

There are two points that are not clear to me.

1) The first is the disagreement with previous ACF measurements with mother machine data for similar traits. The author argues "These cells might experience slightly different environments at different times resulting from thermal fluctuations and their dynamic interaction with their surroundings […] Thus, averaging over many traps erases the dynamics of cellular memory". I find it hard to imagine that significant variation in tiny microfluidic traps that are flushed actively by fresh medium. In principle, this could be easily checked by performing ACF for each trap (rather than averaging over many traps)?

2) Figure 3C. What is the dashed line for? What does it means that the correlation drops below the dash line immediately within a one generation, before it slowly goes back?

Reviewer #3:

This is an interesting paper introducing a microfluidics system, the "sister machine", that traps two *Escherichia coli* sister cells in a v-shaped ending, which allows to study the progeny of each sister over future generations. This is a novel approach and the device, in fact, allows certain measurements that were not possible with previous microfluidics systems. The widely-used mother machine traps one mother cell in each channel, hence the sister cell is eventually pushed away from the channel within several generations.

With this new device, authors performed live cells microscopy experiments and proceeded with detailed image and statistical analysis of individual cells in three different pairing categories: (1) Sister cells (SCs), (2) Neighbor cells (NCs), and (3) Random cell pairs (Rps). The sister cells are form the same mother cells. The neighbor cells are from random mothers but positioned next to each other similar to two sister cells. And the random pairs are cells from random mothers in different channels. Comparison between these three categories reveals long-term memory in specific cellular characteristic (size, generation time, growth rate, protein expression, etc) that may arise only in sister cells progenies.

The quantities that were measured in this work are (1) cell size, (2) generation time, (3) elongation rate, (4) fluorescent intensity (a proxy for protein expression). Authors presented Pearson- and auto-correlation of each pairing category to show there are long-term effects (up to ~9 generations) among sister cells that are not found in random pairs or non-sister neighbor cells.

Overall, this is an interesting manuscript mainly due to the introduction of the v-shaped microfluidics device and I do recommend it for publication. However, I am concerned that the focus of the results in the main text are on the cells size and generation time, which are somehow predictable from our prior knowledge on cell size (as I explain below in 3e). As such, the data does not fully reveal the potential of the device. Perhaps focus on protein expression data is a better choice for demonstrating sister cell correlations which in fact show distinct feature as depicted in Figure 3C.

Below I have detailed point-by-point comments. Most of the items blow concerned with presentations and are rather suggestions.

1) Abstract

a) The authors mention that they "introduce a new experimental method" and discuss the findings and results of using the device. But unfortunately there is not mentioned of the new experimental method, which I think is the single most important part of this work. I suggest including a short description of sisters cells trapping early in the Abstract.

b) I also suggest adding a few words to elucidate "physical and functional characteristics" term in the Abstract.

2) Introduction

a) The opening paragraph contains many terms that need to defined or elucidated such as "physical and functional characteristics", "cellular characteristics", "the state of the cell", and "non-genetic cellular components."

b) The statement "… non-genetic memory in bacteria is almost completely erased within one generation" does not apply to every parameter. Cell-size is known to have a long-term memory as authors discuss later.

c) Because of the small size, this is not clear how much "different environment" cells may experience in a microfluidics system. Thus, the statement "These cells might experience slightly different environments at different times resulting from.…", needs to be backed by data or references.

d) Again, more important than the environmental fluctuation, the significance of the device is that it allows a certain tracking of the lineage which was not possible with mother machine. I believe this needs to be emphasized more clearly.

3) Results

a) Is there any side-effects from physical confinement and bending of the mother cell at the v-shaped end? This needs detailed discussion.

b) Since number of generations and imaging interval is mentioned, it helps if authors add the cells' average generation time too. The explanation of NCs and the difference between them and SCs is not very clear in the text. The figure made it clear though.

c) "Since NCs and RPs do not originate from the same mother at time 0, the PCF is measured from the first generation only, and we set it to be 1 at time 0." Why is the correlation set to 1? If they are from different mother cells, isn't that expected to be zero?

d) Figure 2 has number of issues.

– With no panel title it is not clear what quantity is plotted without reading the caption.

– Axes limit need adjustment. Some error bars span outside the border.

– The variable “g” is not defined. Is it the generation number?

– The grid lines are not necessary.

e) The *E. coli* cell-size is known to have long-term memory based on the adder model. Thus, the progeny of each sister cell will retain the correlation with the mother cell for a number of generations. Thus it is not a surprise to see stronger Pearson correlation between the progenies of a sister pair that random pairs. The same goes with the generation time. Perhaps similar data for protein expression shows something that has not been studied before?

f) Presentation of all panels of Figure 3 can be improved by eliminating grid lines and using a solid line with shades to depict error bars.

Reviewer #4:

The paper by Vashistha et al. presents a new method to investigate the inheritance and memory of cellular characteristics in bacteria, such as cell cycle properties (cell size, division time…), protein content or growth rate. This method is based on the development of a new microfluidic device, that the authors name the "sisters machine", that allows keeping sister cells created from a single mother close to one another in a v-shaped channel, for tens of generation. These sister cells and their descendants therefore share the same microenvironment. The authors can then compute the correlation between the characteristics of the two sister cells and their progeny as a function of time. They can then compare this correlation function to the same correlation obtained on non-related cells sharing the same microenvironment, or on non-related cells in different channels/microenvironment. In doing so they aim at disentangling the effect of epigenetic memory and micro-environment fluctuations.

I think introducing a new tool to disentangle environment fluctuations and epigenetic memory is very interesting, and this new microfluidic chip offers great possibilities in that regard. Given the small dimensions of the chip and the precision that is probably needed for the tip of the v-shaped channels (so that sister cells can be trapped for generations), the development of this sisters machine represents an impressive technical achievement. The data produced in this work is new and very interesting. However, I think this paper does not present the theoretical foundations that are required to interpret the results and quantify inheritance of cellular characteristics. I will list below several issues that are completely unclear for me and that would probably all be resolved by defining a clear theoretical framework.

– I do not understand why the autocorrelation functions (ACF) vanish so fast (Figure 2B and C). The authors say (l.60-61) that averaging over many channels erases the memory. But I do not see why/how this would be the case. I believe there are several models in which this is not true. For simplicity let's take a non-normalized form for the ACF of X(t), i.e. E(X(t)X(t+tau)) and assume all the variables are of average 0. Let's say that the random variable of interest at time t X(t) (for example cell cycle time) is the sum of an environment-dependent noise N(t) and an environment-independent variable Y(t). Then the ACF of X is the sum of the ACF of N(t) and the ACF of Y(t) (assuming independence of Y and N). So the inheritance of environmental fluctuations and the inheritance of epigenetic fluctuations are entangled, but the ACF does not vanish.

– From the definition of the PCF (Equation 1) I can see clearly why the PCF decreases with time for SC but I cannot understand why it is not constant for NC in Figure 2 (the only way I see to have a decreasing function for NC is that the environmental fluctuations are non-stationary, and their variance decreases with time)

– Likewise I understand the trends in Figure 3—figure supplement1 panel D, where the variance is constant for NC and RP and increases for SC. But why is it different in Figure 3—figure supplement 2 ?

To answer these questions and all the others that could be raised by this interesting data, I think a theoretical framework has to be clearly defined. Maybe the authors had such a framework in mind when interpreting the data. In this case I would recommend that they define it clearly in the paper. If on the contrary, the development of such a framework is beyond the scope of this work, then simulations should be provided and compared with the data. Otherwise it is very difficult to interpret the data and demonstrate the validity of the method.

Another important point that was unclear to me : how are NC pairs of cells defined? I understood that they are not sisters but how can the authors be sure that they are not cousins (1st, 2nd, 3rd, 4th cousins…)? In which case the NC cells at time 0 would be the same as the SC cells but at a larger generation (this would explain why the PCF of NC cells decrease in Figure 2). This should be clarified in the text.

[Editors’ note: further revisions were suggested prior to acceptance, as described below.]

Thank you for resubmitting your work entitled "Non-genetic inheritance restraint of cell-to-cell variation" for further consideration by *eLife*. Your revised article has been evaluated by Aleksandra Walczak (Senior Editor) and a Reviewing Editor.

The manuscript has been improved but there are a few small remaining issues that need to be addressed before acceptance. In particular, please note the comments from reviewer 1 regarding the need to clarify the tracking period in the appendix, from reviewer 2 requesting proper citation of relevant articles from the cell size literature, and from reviewer 3 requesting clarification of how your results compare with those of previous studies and a more thorough explanation of the data in Figure 1—figure supplement 1. See full reviewer comments below:

Reviewer #1:

The manuscript by Vashistha et al. is a revised manuscript. My concerns about the previous version were satisfyingly addressed in the present version and in the authors' answers. So I recommend this manuscript for publication

I just have a minor comment :

I now understand that NC cells are tracked from a point where they divide at the same time and have approximately the same lengths (this is now clearly stated in the present version, I hadn't understood from the previous one…). However in the appendix where the mathematical framework is presented (which I find very useful) it is not stated and I think it may be misleading. I think it could be useful to mention that in the appendix too

Reviewer #2:

Since cell size is discussed in the manuscript, I suggest authors to cite related and key publications in the field of cell-size too.

Reviewer #3:

I believe that Figure 1—figure supplement 1 is new. I like this graph, but it requires further explanation. The legend says "This presentation emphasizes the difference in the average growth rates measured in different traps. Note however, that each pair of SCs exhibits similar average growth rate." Can authors quantify the difference and similarity? It is hard to deduce that numbers from the graph. And also, do authors know whether this variation from different traps is also a problem in previous experimental set-up (mother machine)?

---

## [Author Response]

[Editors’ note: the authors resubmitted a revised version of the paper for consideration. What follows is the authors’ response to the first round of review.]

Reviewer #2:This article describes correlation timescale of various cellular properties. The experimental design is ingenious, and findings shed new light on epigenetic inheritance. I believe that this article warrants publication in eLife.There are two points that are not clear to me.1) The first is the disagreement with previous ACF measurements with mother machine data for similar traits. The author argues "These cells might experience slightly different environments at different times resulting from thermal fluctuations and their dynamic interaction with their surroundings […] Thus, averaging over many traps erases the dynamics of cellular memory". I find it hard to imagine that significant variation in tiny microfluidic traps that are flushed actively by fresh medium. In principle, this could be easily checked by performing ACF for each trap (rather than averaging over many traps)?

This is a good point, and we apologize for not including this information before. We have now added a new supplementary figure (Figure 1—figure supplement 1B) that depicts single traps ACFs and their average. As can be seen in the figure, each trap exhibits different behavior with distinct ACF, and the average ACF decays exponentially with a decay time of 2 generations. We address this in the text that describes Figure 2, where we also cite 2 other references that have presented similar calculations with the same result, namely references: Tanouchi et al., 2015, and Susman et al., 2018.

2) Figure 3C. What is the dashed line for? What does it means that the correlation drops below the dash line immediately within a one generation, before it slowly goes back?

We think that the reviewer means the dashed line in Figure 3B. In that case, the line represents the average variance (not correlation) measured for the NCs and RPs depicted in Figure 3A. The fact that the variance of SCs is higher at time zero and drops immediately below that line within one generation before it slowly goes back up reflects that immediately after separation, the sisters exhibit large growth rate variance (the variance is actually larger than the variance exhibited by unrelated cells, i.e. NCs and RPs represented by the dashed line), which means lower correlation. However, they become very similar towards the end of the first cell cycle, i.e. more correlated. After the first cell cycle, the growth rates of the SCs start to diverge again but very slowly until they exhibit similar variance to unrelated cells after ~7 generations. We explain this now in the main text addressing the figure. This is discussed in the second paragraph before the Discussion section.

Reviewer #3:This is an interesting paper introducing a microfluidics system, the "sister machine", that traps two *Escherichia coli* sister cells in a v-shaped ending, which allows to study the progeny of each sister over future generations. This is a novel approach and the device, in fact, allows certain measurements that were not possible with previous microfluidics systems. The widely-used mother machine traps one mother cell in each channel, hence the sister cell is eventually pushed away from the channel within several generations.With this new device, authors performed live cells microscopy experiments and proceeded with detailed image and statistical analysis of individual cells in three different pairing categories: (1) Sister cells (SCs), (2) Neighbor cells (NCs), and (3) Random cell pairs (Rps). The sister cells are form the same mother cells. The neighbor cells are from random mothers but positioned next to each other similar to two sister cells. And the random pairs are cells from random mothers in different channels. Comparison between these three categories reveals long-term memory in specific cellular characteristic (size, generation time, growth rate, protein expression, etc) that may arise only in sister cells progenies.The quantities that were measured in this work are (1) cell size, (2) generation time, (3) elongation rate, (4) fluorescent intensity (a proxy for protein expression). Authors presented Pearson- and auto-correlation of each pairing category to show there are long-term effects (up to ~9 generations) among sister cells that are not found in random pairs or non-sister neighbor cells.Overall, this is an interesting manuscript mainly due to the introduction of the v-shaped microfluidics device and I do recommend it for publication. However, I am concerned that the focus of the results in the main text are on the cells size and generation time, which are somehow predictable from our prior knowledge on cell size (as I explain below in 3e). As such, the data does not fully reveal the potential of the device. Perhaps focus on protein expression data is a better choice for demonstrating sister cell correlations which in fact show distinct feature as depicted in Figure 3C.Below I have detailed point-by-point comments. Most of the items blow concerned with presentations and are rather suggestions.1) Abstracta) The authors mention that they "introduce a new experimental method" and discuss the findings and results of using the device. But unfortunately there is not mentioned of the new experimental method, which I think is the single most important part of this work. I suggest including a short description of sisters cells trapping early in the Abstract.

We have added a short description of the method as suggested.

b) I also suggest adding a few words to elucidate "physical and functional characteristics" term in the Abstract.

We have explained “physical and functional characteristics” better now.

2) Introductiona) The opening paragraph contains many terms that need to defined or elucidated such as "physical and functional characteristics", "cellular characteristics", "the state of the cell", and "non-genetic cellular components."

We have now added definitions for all these terms.

b) The statement "… non-genetic memory in bacteria is almost completely erased within one generation" does not apply to every parameter. Cell-size is known to have a long-term memory as authors discuss later.

Previous measurements of cell-size autocorrelation function by the mother machine has consistently showed no memory, as seen in all the references we cite there including Wang et al., 2010, Tanouchi et al., 2015, and Susman et al., 2018. Indeed, our measurements do show that there is a long memory including in cell-size, however, this is a new result, and the statement the reviewer is referring to here is meant to convey what is the current state of the measurements available in order to emphasize the importance of our new results presented here.

c) Because of the small size, this is not clear how much "different environment" cells may experience in a microfluidics system. Thus, the statement "These cells might experience slightly different environments at different times resulting from.…", needs to be backed by data or references.

We have explained this better now, and added in addition to the references, a supplementary figure (Figure 1—figure supplement 1B), which shows the distinct ACFs measured in different traps, and their average that shows no memory.

d) Again, more important than the environmental fluctuation, the significance of the device is that it allows a certain tracking of the lineage which was not possible with mother machine. I believe this needs to be emphasized more clearly.

We emphasize now that we do track two lineages of sister cells simultaneously using this technique for tens of generations.

3) Resultsa) Is there any side-effects from physical confinement and bending of the mother cell at the v-shaped end? This needs detailed discussion.

We thank the reviewer for reminding us of this important test. We now added a new supplementary figure (Figure 1—figure supplement 2) that compares the effect of division, in our V-shaped device to that in the mother machine, on different parameters. Our results show that there is almost no distinct difference between division in the straight traps of the mother machine and in the V-shaped ones. The largest difference is observed in the growth rate, where the division in the mother machine exhibits larger variation between the two cells than in our device. This could be due to the fact that in the mother machine, the daughter cell is always closer to the food source (diffusing from the flow channel into the trap) than its mother, whereas in the V-Shaped device, both cells are at the same distance from the food source. Therefore, the similarity in our device between the two cells immediately after division is either the same or larger than in the mother machine as expected.

b) Since number of generations and imaging interval is mentioned, it helps if authors add the cells' average generation time too. The explanation of NCs and the difference between them and SCs is not very clear in the text. The figure made it clear though.

The average generation time is now provided in the same paragraph. And the difference between NCs and SCs is clarified.

c) "Since NCs and RPs do not originate from the same mother at time 0, the PCF is measured from the first generation only, and we set it to be 1 at time 0." Why is the correlation set to 1? If they are from different mother cells, isn't that expected to be zero?

This is a good question that we should clarify more. We do that for reasons of comparing the decay time of the correlation between NCs with that of SCs. We choose the NCs such that their size at time 0 is almost identical similar to SCs, and setting the correlation at time 0 to 1, is like assuming that NCs originate from a single mother but they do not share the same epigenetic information. This comparison actually emphasizes that the correlation we measure between SCs is not due to the fact that they have similar size at time 0 like NCs, but rather due to the fact that they share epigenetic information. We explain this point more clearly now in the text.

d) Figure 2 has number of issues.– With no panel title it is not clear what quantity is plotted without reading the caption.– Axes limit need adjustment. Some error bars span outside the border.– The variable “g” is not defined. Is it the generation number?– The grid lines are not necessary.

All issues have been fixed.

e) The *E. coli* cell-size is known to have long-term memory based on the adder model. Thus, the progeny of each sister cell will retain the correlation with the mother cell for a number of generations. Thus it is not a surprise to see stronger Pearson correlation between the progenies of a sister pair that random pairs. The same goes with the generation time. Perhaps similar data for protein expression shows something that has not been studied before?

The prediction of the adder model is that the correlation time of size is ~2 generations. We now add two citations that show this calculation. Namely: Susman et al., 2018, and Ho et al., 2018. We also would like to point out that all measurements of the ACF for size decays over ~2 generation. This is much smaller than what our measurements show, and that is why we consider this result to be significant. We hope that the reviewer will agree to reconsider this point in light of the references we provide here.

f) Presentation of all panels of Figure 3 can be improved by eliminating grid lines and using a solid line with shades to depict error bars.

The suggestions have been included.

Reviewer #4:The paper by Vashistha et al. presents a new method to investigate the inheritance and memory of cellular characteristics in bacteria, such as cell cycle properties (cell size, division time…), protein content or growth rate. This method is based on the development of a new microfluidic device, that the authors name the "sisters machine", that allows keeping sister cells created from a single mother close to one another in a v-shaped channel, for tens of generation. These sister cells and their descendants therefore share the same microenvironment. The authors can then compute the correlation between the characteristics of the two sister cells and their progeny as a function of time. They can then compare this correlation function to the same correlation obtained on non-related cells sharing the same microenvironment, or on non-related cells in different channels/microenvironment. In doing so they aim at disentangling the effect of epigenetic memory and micro-environment fluctuations.I think introducing a new tool to disentangle environment fluctuations and epigenetic memory is very interesting, and this new microfluidic chip offers great possibilities in that regard. Given the small dimensions of the chip and the precision that is probably needed for the tip of the v-shaped channels (so that sister cells can be trapped for generations), the development of this sisters machine represents an impressive technical achievement. The data produced in this work is new and very interesting. However, I think this paper does not present the theoretical foundations that are required to interpret the results and quantify inheritance of cellular characteristics. I will list below several issues that are completely unclear for me and that would probably all be resolved by defining a clear theoretical framework.– I do not understand why the autocorrelation functions (ACF) vanish so fast (Figure 2B and C). The authors say (l.60-61) that averaging over many channels erases the memory. But I do not see why/how this would be the case. I believe there are several models in which this is not true. For simplicity let's take a non-normalized form for the ACF of X(t), i.e. E(X(t)X(t+tau)) and assume all the variables are of average 0. Let's say that the random variable of interest at time t X(t) (for example cell cycle time) is the sum of an environment-dependent noise N(t) and an environment-independent variable Y(t). Then the ACF of X is the sum of the ACF of N(t) and the ACF of Y(t) (assuming independence of Y and N). So the inheritance of environmental fluctuations and the inheritance of epigenetic fluctuations are entangled, but the ACF does not vanish.

This is an excellent point, which we now try to explain better in the manuscript. In short, fluctuations in the environmental conditions do influence the cell’s internal state such as growth rate, and in the mother machine or any microfluidic trap with such small dimensions, fluctuations in the growth rate can influence the concentration of nutrients in the cell’s surroundings. Therefore, the Y and N in the reviewer comment are not independent of each other. Their interaction though is distinct in different traps. This is what we observe in the ACF calculation, which we now present in Figure 1—figure supplement 1.

– From the definition of the PCF (Equation 1) I can see clearly why the PCF decreases with time for SC but I cannot understand why it is not constant for NC in Figure 2 (the only way I see to have a decreasing function for NC is that the environmental fluctuations are non-stationary, and their variance decreases with time)– Likewise I understand the trends in Figure 3—figure supplement1 panel D, where the variance is constant for NC and RP and increases for SC. But why is it different in Figure 3—figure supplement 2 ?

We explain the NCs now better. The reviewer is right in expecting the NCs’ PCF to be zero in general, and now we add a supplemental figure to show that (Figure 2—figure supplement 5). However, as we try to explain now in reference to Figure 2, the NCs we use for comparison are cells that, at time zero, have size difference as small as that observed in SCs. The point of this choice is to emphasize that the long-term correlation observed between SCs is not simply due to the fact that they are similar in size and reside in the same environment, which is the case of NCs, but rather because they share epigenetic memory. If the observed long-term correlations in cell size between SCs was just due to the fact that they start from similar size at time zero and that they reside in the same environment, then the same correlation would be observed between NCs that start from a similar size.

To answer these questions and all the others that could be raised by this interesting data, I think a theoretical framework has to be clearly defined. Maybe the authors had such a framework in mind when interpreting the data. In this case I would recommend that they define it clearly in the paper. If on the contrary, the development of such a framework is beyond the scope of this work, then simulations should be provided and compared with the data. Otherwise it is very difficult to interpret the data and demonstrate the validity of the method.

We have added a mathematical appendix explaining the relationships between the different measures we use and how they relate to the nongenetic memory and restrain of variation. A more detailed explanation and a quantitative extraction of further information requires much more space and is beyond the scope of this study. We hope however, that the reviewer finds this additional information that we provide here sufficient to frame the problem for the readers and help them interpret the data properly.

Another important point that was unclear to me : how are NC pairs of cells defined? I understood that they are not sisters but how can the authors be sure that they are not cousins (1st, 2nd, 3rd, 4th cousins…)? In which case the NC cells at time 0 would be the same as the SC cells but at a larger generation (this would explain why the PCF of NC cells decrease in Figure 2). This should be clarified in the text.

We do know that these cells are not closely related because they are usually in the V-shaped channel from the start of the experiment, and we track them only from a point where their division occurs simultaneously. This usually happens after few divisions. Although some of the cells initially trapped in the channels can be indeed closely related, statistically this has a very low probability.

[Editors’ note: what follows is the authors’ response to the second round of review.]

Reviewer #1:The manuscript by Vashistha et al. is a revised manuscript. My concerns about the previous version were satisfyingly addressed in the present version and in the authors' answers. So I recommend this manuscript for publicationI just have a minor comment :I now understand that NC cells are tracked from a point where they divide at the same time and have approximately the same lengths (this is now clearly stated in the present version, I hadn't understood from the previous one…). However in the appendix where the mathematical framework is presented (which I find very useful) it is not stated and I think it may be misleading. I think it could be useful to mention that in the appendix too

We have added this text as requested in the appendix.

Reviewer #2:Since cell size is discussed in the manuscript, I suggest authors to cite related and key publications in the field of cell-size too.

We have added citations of few important papers in the field of cell size homeostasis in the Introduction, as well as other studies of the growth rate in connection to size homeostasis in the Results section.

Reviewer #3:I believe that Figure 1—figure supplement 1 is new. I like this graph, but it requires further explanation. The legend says "This presentation emphasizes the difference in the average growth rates measured in different traps. Note however, that each pair of SCs exhibits similar average growth rate." Can authors quantify the difference and similarity? It is hard to deduce that numbers from the graph.

We thank the reviewer for this important comment. We have now changed the figure and added a new figure, which presents the distributions of growth rates differences between sister cells and between random pairs. The distributions clearly show that the standard deviation of differences between sisters is almost half of that of random pairs occupying different traps. The old Figure 1-Figure 1A is now Figure 2A in the new format of the manuscript, and the new figure is Figure 2B, while the old Figure 1—figure supplement 1B is now Figure 1—figure supplement 1.

And also, do authors know whether this variation from different traps is also a problem in previous experimental set-up (mother machine)?

We think that this did occur in previous studies based on our own analysis of data published by other groups (see for example Susman et al., 2018). However, for the purpose of those studies, we don’t think that this effect changes their conclusions, since there was no distinction there between environmental fluctuations and molecular ones for the questions they were trying to address.